# Smooth and Flexible Camera Movement Synthesis via Temporal Masked Generative Modeling

**Chenghao Xu[1], Guangtao Lyu[1], Jiexi Yan[2*], Muli Yang[3], Cheng Deng[1*]**
[1] School of Electronic Engineering, Xidian University, Xi'an, Shaanxi, China,
[2] School of Computer Science and Technology, Xidian University, Xi'an, Shaanxi, China,
[3] Institute for Infocomm Research (I$^2$R), A*STAR, Singapore
{chx,guangtaolyu}@stu.xidian.edu.cn,
{jxyan1995,muliyang.xd,chdeng.xd}@gmail.com

## Abstract

In dance performances, choreographers define the visual expression of movement, while cinematographers shape its final presentation through camera work. Consequently, the synthesis of camera movements informed by both music and dance has garnered increasing research interest. While recent advancements have led to notable progress in this area, existing methods predominantly operate in an offline manner—that is, they require access to the entire dance sequence before generating corresponding camera motions. This constraint renders them impractical for real-time applications, particularly in live stage performances, where immediate responsiveness is essential. To address this limitation, we introduce a more practical yet challenging task: online camera movement synthesis, in which camera trajectories must be generated using only the current and preceding segments of dance and music. In this paper, we propose TemMEGA (Temporal Masked Generative Modeling), a unified framework capable of handling both online and offline camera movement generation. TemMEGA consists of three key components. First, a discrete camera tokenizer encodes camera motions as discrete tokens via a discrete quantization scheme. Second, a consecutive memory encoder captures historical context by jointly modeling long- and short-term temporal dependencies across dance and music sequences. Finally, a temporal conditional masked transformer is employed to predict future camera motions by leveraging masked token prediction. Extensive experimental evaluations demonstrate the effectiveness of our TemMEGA, highlighting its superiority in both online and offline camera movement synthesis.

## 1 Introduction

Recent advances in image generation have significantly enhanced visual storytelling in performance arts [26; 43; 42]. In dance performances, camera work is pivotal in shaping the audience's perception and interpretation of the choreography [28; 30; 39; 3; 46]. By employing multiple camera angles and transitions, producers can better capture key dance movements, offering a more immersive storytelling experience. Additionally, creative techniques such as quick cuts, slow motion, and dolly shots enhance visual impact and introduce novelty, thereby increasing the performance's overall appeal. However, the movement of the camera is influenced by several factors, including the music and the choreography itself. Moreover, effective dance cinematography requires a variety of shot types and a focus on human-centered elements. As a result, the automatic generation of camera movements based on music and dance remains a compelling yet complex challenge.

---

*Corresponding author

39th Conference on Neural Information Processing Systems (NeurIPS 2025).

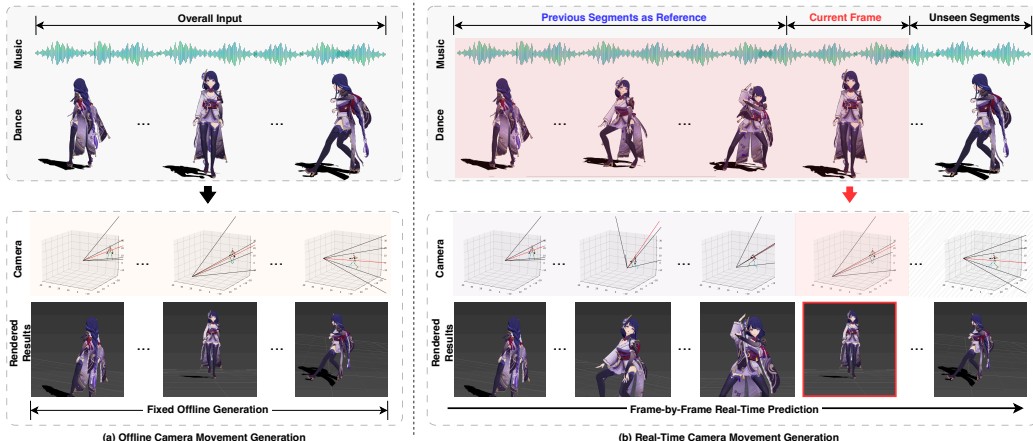

Figure 1: **Illustration of the traditional offline camera movement generation task and our proposed real-time camera movement synthesis.** (a) Offline Camera movement generation: The entire dance video with music is available to synthesize the corresponding camera movements; (b) Online camera movement generation: Camera movements are generated frame by frame. For the current frame, we employ the previous and current segments of dance and music as input to synthesize the corresponding camera movement.

Previously, significant attention had been given to camera planning and control [27; 45], primarily in gaming and film scenes. Recently, several methods have aimed to tackle the more challenging task of dance camera synthesis. Among these, DanceCamera3D [37] introduced the first 3D dance-camera-music dataset (DCM) and demonstrated the feasibility of synthesizing camera movements driven by music and dance. Additionally, Cine-AI [40] simplifies the problem by reducing it from 3D to 2D, excluding the camera's roll and pitch orientation. This simplification significantly diminishes the expressiveness of the camera movements and reduces the complexity of the task. Moreover, DanceCamAnimator [38] integrates human animation knowledge into the problem of music- and dance-driven camera synthesis, employing this knowledge to generate 3D camera movements by following animators' hierarchical camera-making procedures.

Although significant progress [7; 9; 12; 14; 15; 18; 19; 4] has been achieved in camera movement synthesis, a major limitation persists in the real-world application of these offline methods due to their reliance on having access to the entire dance video as input, as shown in Figure 1(a). In practice, online requirements must be met in the camera movement generation process, meaning that camera movements need to be swiftly generated during live stage performances, where the complete dance video is not available; instead, only prior segments of the performance are accessible. Consequently, we focus on a more practical yet challenging task, namely, online camera movement synthesis illustrated in Figure 1(b).

In this paper, we introduce TemMEGA, a novel Temporal Masked Generative Modeling framework for both online and offline camera movement synthesis. Our approach is built upon three key components. First, the discrete camera tokenizer (DCT) is trained using the vector quantized variational autoencoder (VQ-VAE). The DCT transforms and quantizes raw camera movement data into a sequence of discrete motion tokens in latent space, based on a camera codebook. To more effectively capture the temporal context of dance and music segments, we introduce the consecutive memory encoder (CME), which provides a more accurate history summary by jointly modeling long- and short-term temporal memories. Specifically, long- and short-term segments of dance motions and music are encoded into fixed tokens. Finally, we mask the tokens to be predicted and employ a conditional masked transformer (CMT) to predict the masked tokens in real-time, conditioned on both the unmasked tokens and the long- and short-term memory. Extensive comparative and ablation studies on public datasets validate the effectiveness of our framework.

In summary, our main contributions include:

- We introduce the practical task of online camera movement synthesis, with the potential to significantly expand applications of camera movement generation, particularly in live stage performances.

- We propose a novel temporal masked generative modeling framework, TemMEGA for smooth and flexible generate camera movement synthesis in both online and offline manner. Our TemMEGA consists of three main components, *i.e.* discrete camera tokenizer, consecutive memory encoder, and conditional masked transformer.

- Comprehensive experiments on public datasets demonstrate that our method achieves state-of-the-art performance, confirming its effectiveness.

## 2 Related Work

### 2.1 Camera Control and Planning

Automatic cinematography has gained significant attention due to the expertise and labor required for manually producing film-like videos, despite the importance of artistic video content in media, entertainment, and gaming industries. Jiang *et al*. [18] propose extracting camera behaviors from film clips for re-application in virtual environments. Similarly, Rao *et al*. [28] generate dynamic storyboards from story and camera scripts, while Wu *et al*. [39] develop a GAN-based controller to produce actor-driven camera movements considering spatial, emotional, and aesthetic factors. Rucks *et al*. [30] introduce CamerAI to replicate chase camera techniques in third-person games, and Evin *et al*. [7] present Cine-AI to simulate movie directors' cinematographic techniques for enhancing game cutscenes. In the domain of aerial cinematography, studies [14; 16; 13; 10] focus on automating drone camera movements based on artistic principles. However, controlling cameras for dance sequences is more complex due to the need to synchronize with music and dance motions.

To address this, Wang et al. [37] introduced the 3D dance-camera-music dataset DCM and developed DanceCamera3D, a transformer-based diffusion model for dance camera synthesis. Nonetheless, it overlooks the mix of continuous shots and abrupt transitions in dance cinematography. DanceCamAnimator [38] improves on this by integrating animator knowledge into a three-stage process—keyframe detection, keyframe synthesis, and tween function prediction—offering precise control over variable-length sequences.

### 2.2 Dance Synthesis

Music-conditioned 3D dance generation merges dance and machine learning, producing dance sequences that align with music's melody and rhythm. Existing approaches are split into two types: retrieval-based and direct generation methods. Retrieval-based approaches [24; 8] segment dances into fixed-length pieces to match the music structure, but are limited by BPM and fixed segment lengths, making synchronization challenging. Direct generation methods [1; 32; 33; 41] address these limitations by generating dance movements from scratch.

Recent advances in deep learning have led to the rise of diffusion-based and discrete generation techniques. Diffusion models, known for their noise-refinement process, generate coherent dance sequences aligned with musical cues. For instance, EDGE [33] employs conditional diffusion models to create dance movements using Jukebox [6] for audio feature extraction. Discrete generation follows a two-stage process. First, VQ-VAE [35] transforms dance movements into compact, discrete features. Next, natural language processing techniques, such as autoregressive and mask modeling, generate and reconstruct dance sequences, ensuring temporal coherence and fluidity while synchronizing with the music.

## 3 Method

### 3.1 Problem Formulation

The existing music-dance-to-camera synthesis methods [37; 38] take a dance video with $T$ frames of music features $\mathcal{A} = \{\mathbf{a}_1, \mathbf{a}_2, \cdots, \mathbf{a}_T\}$ and dance motions $\mathcal{M} = \{\mathbf{m}_1, \mathbf{m}_2, \cdots, \mathbf{m}_T\}$ as input conditions, to generate camera movement sequence $\mathcal{C} = \{\mathbf{c}_1, \mathbf{c}_2, \cdots, \mathbf{c}_T\}$, which is a offline paradigm.

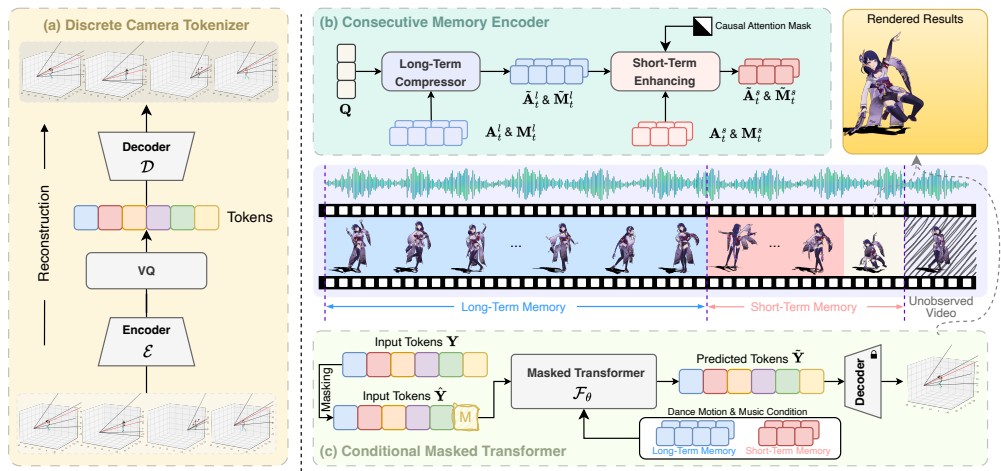

Figure 2: **The framework of our proposed TemMEGA.** Our method can be mainly divided in three components, *i.e.* discrete camera tokenizer (DCT), consecutive memory encoder (CME) and conditional masked transformer (CMT).

Considering the more practical setting, *i.e.* online music-dance-to-camera synthesis as illustrated in Figure 1(b), we generate the current camera movement $\mathbf{c}_t$ by taking the current and previous $t$ frames of music features $\mathcal{A}_t = \{\mathbf{a}_1, \mathbf{a}_2, \cdots, \mathbf{a}_t\}$ and dance motions $\mathcal{M}_t = \{\mathbf{m}_1, \mathbf{m}_2, \cdots, \mathbf{m}_t\}$.

Specifically, we follow FACT [21] to extract music features, denoted as $m_t \in \mathbb{R}^{35}$, using Librosa [23]. For dance motions and camera movements, we adopt the approach of DanceCamera3D [37], using the global positions of 60 human joints, represented as $\mathbf{m}_i \in \mathbb{R}^{60 \times 3}$, and MMD format camera representation in polar coordinates, denoted as $\mathbf{c}_i \in \mathbb{R}^{3+3+1+1}$. This includes the global position of the reference point, the camera's rotation and distance relative to the reference point, and the camera's field of view (FOV).

## 3.2 Temporal Masked Generative Modeling

Our objective is to develop a unfied solution for both offline and online music-dance-to-camera synthesis that efficiently generates camera movements in real time by utilizing previous and current dance and music segments. To accomplish this, we propose a novel temporal masked generative modeling (TemMEGA) framework to replace previous diffusion-based methods, which are limited to offline operation, i.e., relying on the entire dance video as input to generate the corresponding camera movement sequence.

As illustrated in Figure 2, our framework consists of three key components. First, the Discrete Camera Tokenizer (DCT) is designed to transform camera movements into a sequence of discrete camera tokens while preserving rich correlated information about the camera movements. Second, the Consecutive Memory Encoder (CME) enhances the conditional information (previous and current music and dance motion) and provides a more accurate history summary of the temporal condition by compressing long- and short-term memory in a segment-based manner. Finally, the Temporal Conditional Masked Transformer (CMT) is trained to predict masked current camera tokens based on the pre-computed long- and short-term memories of both music and dance motion.

**Discrete Camera Tokenizer.** To effectively facilitate the synthesis of camera movements, we pre-train a discrete camera tokenizer (DCT). This is achieved using the Vector Quantized Variational Autoencoder (VQ-VAE) architecture [35; 41], which enables the generation of discrete representations of camera shot data through the quantization of encoder outputs into discrete tokens, mapped to entries or codes from a learned codebook via vector quantization. Our DCT framework comprises a camera encoder $\mathcal{E}$ and a camera decoder $\mathcal{D}$. The objective of vector quantization is defined as follows:

$$\mathcal{L}_{VQ} = ||\text{sg}[\mathcal{E}(\mathbf{c}_i)] - \hat{\boldsymbol{\nu}}_i||_2^2 + \beta||\mathcal{E}(\mathbf{c}_i) - \text{sg}[\hat{\boldsymbol{\nu}}_i]||_2^2. \tag{1}$$

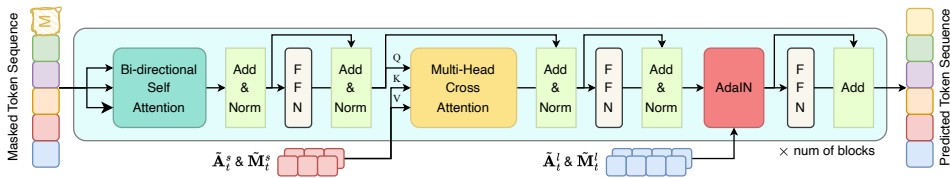

Figure 3: **Architecture of the masked transformer.** The model is a multi-layer transformer with a bidirectional attention structure. It takes as input the masked camera tokens, along with the long- and short-term memories of music and dance motion, to predict the camera tokens. The long- and short-term memories of music and dance motion are integrated into the network at various stages through self-attention layers and AdaIN layers, respectively.

Here, for the $i$-th latent feature $\boldsymbol{\nu}_i$, the estimated embedding $\hat{\boldsymbol{\nu}}_i$ can be found by searching the nearest embedding in the codebook $\mathcal{X}$ through the quantization process $Q(\cdot)$:

$$\hat{\boldsymbol{\nu}}_i = Q(\boldsymbol{c}_i) := \arg\min_{\boldsymbol{x}_k \in \mathcal{X}} \|\boldsymbol{\nu}_i - \boldsymbol{x}_k\|_2. \tag{2}$$

Based on the estimation latent representation $\hat{\boldsymbol{V}} = [\hat{\boldsymbol{\nu}}_1, \hat{\boldsymbol{\nu}}_2, \cdots, \hat{\boldsymbol{\nu}}_T]$, the reconstructed camera movements can be produced by the decoder $\boldsymbol{D}(\cdot)$, *i.e.*, $\tilde{\boldsymbol{X}} = \boldsymbol{D}(\hat{\boldsymbol{V}})$.

Additionally, we incorporate moving averages during codebook updates and reset inactive codebooks, techniques commonly used to improve codebook utilization in VQ-VAE. These strategies enable the robust and efficient transformation of camera movements into a sequence of discrete camera tokens.

**Consecutive Memory Encoder.** To more effectively handle the previous conditional information, we draw inspiration from some existing works [44; 36] and introduce a Consecutive Memory Encoder (CME) to separate the long- and short-term memories of the preceding music and dance motion segments. This allows for modeling short-term context while extracting meaningful correlations from the long-term history. By doing so, we compress the long-term history without losing important fine-grained details.

As illustrated in Figure 2(b), we explicitly divide the previous music and dance motion segments into long- and short-term memories. Specifically, for the prediction of the $t$-th frame, the short-term memory retains only a limited number of recent frames of music and dance motion, denoted as $\mathbf{A}_t^s = \{\mathbf{a}_i\}_{i=t-L_s+1}^{t}$ and $\mathbf{M}_t^s = \{\mathbf{m}_i\}_{i=t-L_s+1}^{t}$, respectively. Here, $L_s$ represents the length of the short-term memory. The other memory, referred to as long-term memory, stores features from frames further removed from the current time. It is defined as $\mathbf{A}_t^l = \{\mathbf{a}_i\}_{i=t-L_s-L_l+1}^{t-L_s}$ and $\mathbf{M}_t^l = \{\mathbf{m}_i\}_{i=t-L_s-L_l+1}^{t-L_s}$, where $L_l$ denotes the length of the long-term memory, which is significantly longer than the short-term memory.

To further improve the quality of the compressed long-term memory and enhance the short-term memory, we compress and abstract the long-term memory into a fixed-length latent representation, which is then integrated into the short-term memory. Specifically, we first divide $\mathbf{A}_t^l$ and $\mathbf{M}_t^l$ into non-overlapping memory segments. Next, we apply a weight-shared transformer decoder block with $K$ learnable tokens as the long-term memory queries to query each segment. Through this process, the memory segments are transformed into $K$ segment-level abstract features. Each feature is then average-pooled into a single vector, and these vectors are concatenated to form the compressed long-term segmented memory. Finally, we input the concatenated vectors into two transformer encoder blocks to obtain the final compressed long-term memories, $\tilde{\mathbf{A}}_t^l$ and $\tilde{\mathbf{M}}_t^l$.

To further enhance the short-term memory $\mathbf{A}_t^s$ and $\mathbf{M}_t^s$, we utilize it as a query to retrieve relevant context from the compressed long-term memory. A transformer causal decoder block is employed to aggregate the compressed long-term memory into the short-term memory $\tilde{\mathbf{A}}_t^s$ and $\tilde{\mathbf{M}}_t^s$. The resulting compressed long-term memory, along with the enhanced short-term memory, is then fed into the subsequent Temporal Conditional Masked Transformer (CMT) as conditional input.

**Temporal Conditional Masked Transformer.** As shown in Figure 2(c), we design a bidirectional masked transformer $\mathcal{F}_\theta$, parameterized by $\theta$, to model the camera tokens. Inspired by MAGE [22], the camera tokens $\mathbf{Y}$ are first obtained by passing the encoder output through a vector quantizer

Table 1: Quantitative results on the DCM dataset in the online setting. The best results are indicated as **Bold**, and the second ones are indicated as Underline. - denotes that the self-comparison is meaningless. * denotes that we retrain and retest the method in the online setting.

| Method | Quality | | Diversity | | Dancer Fidelity | | User Study |
|---|---|---|---|---|---|---|---|
| | $\text{FID}_k \downarrow$ | $\text{FID}_s \downarrow$ | $\text{Dist}_k \uparrow$ | $\text{Dist}_s \uparrow$ | $\text{DMR} \downarrow$ | $\text{LCD} \downarrow$ | TemMEGA WinRate $\uparrow$ |
| GT | - | - | 3.275 | 1.731 | 0.00142 | - | 32.15%±3.07% |
| DanceCamera3D* [37] | 4.634 | 0.761 | 1.488 | 1.109 | 0.0066 | 0.197 | 83.43%±2.36% |
| TemMEGA w/o $\tilde{\mathbf{A}}_t^l \& \tilde{\mathbf{M}}_t^l$ | 4.367 | 0.618 | 1.525 | 1.64 | 0.0045 | 0.180 | 61.36%±1.96% |
| TemMEGA | **4.025** | **0.599** | **1.589** | **1.187** | **0.0035** | **0.177** | - |

Table 2: Quantitative results on the DCM dataset in the offline setting. The best results are indicated as **Bold**, and the second ones are indicated as Underline. - denotes that the self-comparison is meaningless.

| Method | Quality | | Diversity | | Dancer Fidelity | | User Study |
|---|---|---|---|---|---|---|---|
| | $\text{FID}_k \downarrow$ | $\text{FID}_s \downarrow$ | $\text{Dist}_k \uparrow$ | $\text{Dist}_s \uparrow$ | $\text{DMR} \downarrow$ | $\text{LCD} \downarrow$ | TemMEGA WinRate $\uparrow$ |
| GT | - | - | 3.275 | 1.731 | 0.00142 | - | 40.35%±2.62% |
| DanceRevolution [17] | 10.267 | 2.368 | 1.491 | 1.118 | 0.0062 | 0.154 | 88.14%±2.05% |
| FACT [21] | 5.205 | 0.960 | 1.505 | 1.007 | 0.0899 | 0.310 | 85.64%±1.61% |
| DanceCamera3D [37] | 3.749 | 0.280 | 1.631 | 1.326 | 0.0025 | 0.147 | 73.64%±2.67% |
| DanceCamAnimator [38] | 3.453 | 0.268 | **3.140** | 1.293 | 0.0022 | 0.152 | 65.64%±4.54% |
| TemMEGA | **3.237** | **0.255** | 1.961 | **1.347** | **0.0020** | **0.141** | - |

during training. We then randomly mask out a varying fraction of the sequence elements, replacing the tokens with a special [MASK] token. The masked camera token sequence $\hat{\mathbf{Y}}$, along with the long- and short-term memories of music and dance motion $\tilde{\mathbf{A}}_t^l, \tilde{\mathbf{M}}_t^l, \tilde{\mathbf{A}}_t^s, \tilde{\mathbf{M}}_t^s$, serve as the inputs for our bidirectional masked transformer $\mathcal{F}_\theta$. Mathematically, the masked transformer $\mathcal{F}_\theta$ is optimized by minimizing the negative log-likelihood of the target predictions:

$$\mathcal{L}_{\text{CMT}} = \sum -\log \mathcal{F}_\theta(\tilde{\mathbf{Y}}|\hat{\mathbf{Y}}, \tilde{\mathbf{A}}_t^l, \tilde{\mathbf{M}}_t^l, \tilde{\mathbf{A}}_t^s, \tilde{\mathbf{M}}_t^s). \tag{3}$$

To effectively integrate the conditional input, we carefully design the masked transformer, as depicted in Figure 3. Following GestureDiffuCLIP [2], in our masked transformer, we extract long-term memory features by compressing and enhancing them, obtaining a fixed-length token representation. These long-term (historical) features capture macro-level information such as the overall style of the music. To incorporate this non-sequential macro-level information into the generation process, we leverage the style transfer capability of the adaptive instance normalization (AdaIN). In contrast, short-term memory features maintain a direct temporal correspondence with the sequence to be generated. Therefore, to ensure the generated sequence aligns temporally with the input short-term information, it is crucial to establish strong temporal interactions between the short-term memory and the generated sequence. To achieve this, we employ a cross-attention mechanism. Moreover, the bidirectional self-attention mechanism enables the prediction of masked tokens by leveraging context from both directions.

**Inference.** During the inference phase in the online setting, we predict only the result at the current time step $t$. Specifically, we append a [MASK] token following the corresponding camera tokens of the short-term memory and utilize both the long- and short-term memories as conditional inputs to predict the masked token. Finally, the predicted tokens are decoded and projected back to camera sequences through the VQ-VAE decoder. During both training and testing, we apply the [MASK] token operation solely to the position corresponding to time $t$ and directly use the model's output at this position as the predicted result for $t$. Consequently, our model does not involve concepts such as masking ratio or the number of inference steps.

## 4 Experiments

In this section, we evaluate our proposed TemMEGA and analyze its essential characteristics.

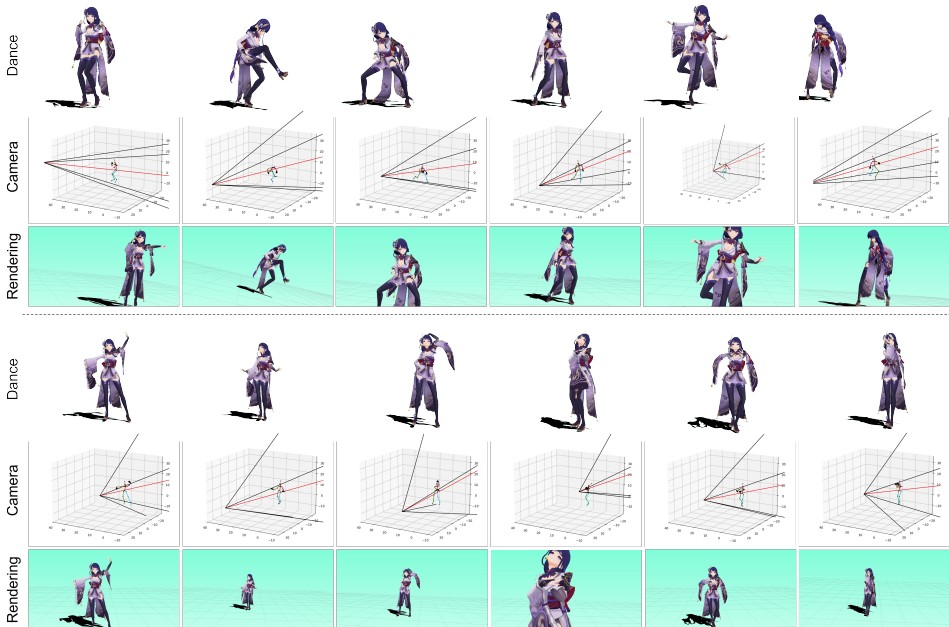

Figure 4: Visualization of the generated results utilizing our TemMEGA.

## 4.1 Datasets and Implementation Details

**Experimental Implementation.** We train our models on 4 NVIDIA A6000 48 GB with a batch size of 512. Discrete Camera Tokenizer (DCT) architecture incorporates residual blocks within its encoder and decoder components, featuring a spatial downscaling factor of 4, which consists of 4 quantization layers, each covering a codebook comprising 2048 vectors of 32-dimensional entities. The quantization dropout ratio is set to 0.2. For Consecutive Memory Encoder (CME), we use two transformer encoder blocks to compress long-term memories with 2 layers, and we enhance the short-term memories with two transformer decoder blocks with 4 layers. we set $L_l$ ,$L_s$ and $K$ to 256, 32 and 8. The number of masked transformer blocks, heads, and dimensions is set to 6, 8, and 512 in the Temporal Conditional Masked Transformer (CMT). We train the models by Adam optimizer [20] with the same hyperparameters (learning rate, $\beta_1$, and $\beta_2$ are set as 0.002, 0, and 0.99, respectively) as previous works.

**Datasets.** In this work, we use DCM [37], a dataset consisting of 108 pieces of animator-designed paired dance-camera-music data including camera keyframe information. To ensure the fairness of the experiment, we follow the previous works and re-use the train and test splits provided by the original dataset. For the training of our framework, in the training set, we stitch the data pieces that are adjacent to the original data so that we acquire more training data with history.

## 4.2 Evaluation Metrics

**Kinetic Feature Evaluation.** Following prior works [21; 31; 37], we evaluate generated camera movement using Fréchet Inception Distance (FID) [11] for quality and average Euclidean distance (Dist) in the feature space for diversity. For kinetic evaluation, we use a kinetic feature extractor [25] following existing works [17; 21; 31]. Since this feature extractor calculates average velocity and acceleration, we compute kinetic features on split 2.5-second data to ensure the density of feature distribution which is similar to settings in AIST++ [21]. Thus, we have got $\text{FID}_k$ for kinetic quality and $\text{Dist}_k$ for kinetic diversity.

**Shot Feature Evaluation.** Shot features play a crucial role in dance camera synthesis. However, existing approaches [5; 29; 34] are confined to 2D classification with a limited number of predefined shot types. Therefore, we use a novel shot feature extractor designed for 3D scenes, incorporating cinematographic knowledge. We follow [37; 38] and calculate shot features as:

$$\text{Features}_{\text{shot}} = (S_3/S_1, S_3/S_2).  \qquad (4)$$

Table 3: Ablation studies on the codebook: (Left) number of code; (Right) code dimension.

| (a) Number of code | | | | |
|---|---|---|---|---|
| noc | $\mathrm{FID}_k$ | $\mathrm{Dist}_k$ | DMR | LCD |
| 256 | 4.302 | 1.498 | 0.0042 | 0.182 |
| 512 | 4.168 | 1.521 | 0.0040 | 0.179 |
| 1024 | 4.088 | 1.563 | 0.0038 | 0.177 |
| 2048 | 4.025 | 1.589 | 0.0035 | 0.177 |
| 4096 | 4.035 | 1.590 | 0.0037 | 0.179 |
| 8192 | 4.125 | 1.568 | 0.0042 | 0.183 |

| (b) Code dimension | | | | |
|---|---|---|---|---|
| cd | $\mathrm{FID}_k$ | $\mathrm{Dist}_k$ | DMR | LCD |
| 16 | 4.091 | 1.561 | 0.0038 | 0.178 |
| 32 | 4.025 | 1.589 | 0.0035 | 0.177 |
| 64 | 4.019 | 1.595 | 0.0034 | 0.176 |
| 128 | 4.015 | 1.605 | 0.0034 | 0.178 |
| 256 | 4.010 | 1.598 | 0.0035 | 0.176 |

where $S_1$ and $S_3$ represent the camera plane projection areas of the dancer's full body and body parts, respectively, within the camera view, and $S_2$ is the total area of the camera screen. The term $S_3/S_1$ indicates the percentage of the body within the camera view, while $S_3/S_2$ reflects the proportion of the camera screen occupied by the dancer. We then compute the Fréchet Inception Distance (FID) and distance (Dist) for Features$_{shot}$ and its velocity to obtain $\mathrm{FID}_s$ and $\mathrm{Dist}_s$, which measure shot quality and diversity. To account for the differences between shot and kinetic features, we calculate shot metrics on a frame-by-frame basis to maintain the accuracy of shot classification.

**Dancer Fidelity Evaluation.** Dancer fidelity means camera movement should try to capture significant body parts against the dancer's poses and avoid the long time absence of the dancer in the camera view. We follow [37; 38] to evaluate dancer fidelity with the following two metrics: **1)** Dancer Missing Rate (DMR): DMR represents the ratio of frames in which the dancer is not in the view of the camera, and **2)** Limbs Capture Difference (LCD): LCD denotes the difference of body parts inside and outside camera view between synthesized results and ground truth. Lower values of DMR and LCD indicate better dancer fidelity, as they correspond to fewer instances where the dancer is missing from the view and greater similarity between the synthesized results and the carefully adjusted ground truth.

**User Study.** For qualitative evaluation, we conduct a user study to compare our method with alternative approaches and the ground truth. In this study, we first randomly select 10 dance-camera input sequences from the test set, each lasting between 17 and 35 seconds. For each sequence, we sample the outputs from our method as well as from baseline methods. This process produces 40 pairs of dance videos, with each pair consisting of the output from our method and one from a baseline method. We then invite 21 participants to view these 30 video pairs in a randomized order and respond to the question, "Which camera movement better highlights the dance and music?" for each pair. The participants include dancers, animators, filmmakers, and individuals with minimal experience in camera work and dance.

### 4.3 Evaluation Results

**Quantitative Results.** We compare our TemMEGA with the state-of-the-art camera generation methods on the DCM dataset and report the experimental results in Table 1. Since DanceCamAnimator [38] is a three-stage approach that requires a complete dance video to generate camera movements, it cannot be applied to real-time generation in the real-time setting. The results indicate that the performance of our proposed method considerably outperforms DanceCamera3D [37]. Notably, our method remains effective even without utilizing long-term memory information. This suggests that both the generation architecture and the proposed Consecutive Memory Encoder play a substantial role in enhancing the quality of the generated results.

To further demonstrate the effectiveness and scalability of our method, we made simple modifications to TemMEGA to enable training and testing in an offline setting. Specifically, we remove the components involving long-term memory in TemMEGA and only use short-term memory as the condition for camera generation. Details of these model modifications can be found in the supplementary materials. We can see that TemMEGA consistently performs favorably against all the other existing methods on all evaluations. The demonstrated superiority of our method across various camera quality metrics indicates that it not only generates motions that are more lifelike compared to those produced by baseline methods, but it also excels in choreographing these movements into coherent camera

Table 4: Ablation studies on temporal receptive field: (Left) local window size $L_l$; (Right) stride $L_s$.

| (a) Impact of $L_l$ | | | | | (b) Impact of $L_s$ | | | | |
|---|---|---|---|---|---|---|---|---|---|
| $L_l$ | $FID_k$ | $Dist_k$ | DMR | LCD | $L_s$ | $FID_k$ | $Dist_k$ | DMR | LCD |
| 64 | 4.112 | 1.554 | 0.0039 | 0.179 | 8 | 4.427 | 1.505 | 0.0054 | 0.192 |
| 128 | 4.068 | 1.575 | 0.0037 | 0.179 | 16 | 4.167 | 1.519 | 0.0047 | 0.186 |
| 256 | 4.025 | 1.589 | 0.0035 | 0.177 | 32 | 4.025 | 1.589 | 0.0035 | 0.177 |
| 512 | 4.017 | 1.593 | 0.0034 | 0.176 | 64 | 4.012 | 1.597 | 0.0033 | 0.175 |
| 1024 | 4.002 | 1.599 | 0.0033 | 0.176 | 128 | 3.995 | 1.602 | 0.0032 | 0.174 |

sequences through the implementation of the proposed CMT, which helps us learn high-fidelity camera.

**Qualitative Results.** To better comprehensively demonstrate the effectiveness of our TemMEGA, we visualize the generated camera shots and corresponding rendered dance with diverse camera movements in Figure 4. Our method demonstrates the ability to achieve smooth dance performances with diverse shot transitions, underscoring the advantages of the TemMEGA framework, which effectively synthesizes satisfactory dance camera movements without requiring the entire dance video. More visual results can be found in the supplementary materials.

## 4.4 Ablation Studies

**The impact for the number of code in the codebook.** We conducted ablation experiments using codebooks of various lengths. Table 3 shows that a codebook length of 2048 yields the best results. When the number of codes in the codebook is reduced, the diversity of the generated outputs diminishes. On the other hand, excessively increasing the number of codes leads to a rapid decline in overall quality. This is because the size of the codebook determines the number of categories in the subsequent CMT classification. When the number of categories becomes too large, it adversely impacts CMT performance.

**The impact of the code dimension of the codebook.** We conducted ablation experiments on codebooks with various dimensions to assess their impact on performance. As shown in Table 3, the results indicate that the code dimension of 32 yields improved outcomes compared to other dimensions. Our experimental analysis suggests that changes in the code dimension have only a minor effect on the quality of generation, indicating relative stability across different dimensions. However, due to the demands of the real-time setting, where high generation speed is essential, we selected a dimension of 32 for the final experiments to balance performance quality with reduced computational cost.

**The choice of $L_l$ and $L_s$** We conducted ablation experiments to assess the effects of different $L_l$ and $L_s$ values. As shown in Table 4, as $L_l$ increases, the generation quality also improves, though this improvement slows when $L_s$ exceeds 256. Table 4 demonstrates that when $L_s$ is smaller, the generation quality increases more rapidly as $L_s$ grows. Given that the real-time setting requires faster generation speeds, we ultimately selected $L_l$ and $L_s$ values of 256 and 32, respectively, as the experimental parameters.

**The choice of $K$** We performed ablation experiments to evaluate the impact of different levels of long-term memory compression, denoted by $K$. As shown in Figure 5, the extent of compression has minimal influence on the outcomes, whereas the inclusion of long-term memory significantly affects performance. Considering the high-speed requirements of the real-time setting, we optimized for a balance between computational efficiency and output quality, selecting $K = 8$ as an effective compromise.

**The impact of components in CME and CMT.** In Table 5, we analyze the influence of different components in CME and CMT. Cases 1–3 correspond to the ablation of CME. Removing the long-term enhancement for short-term encoding (Case 1) increases $FID_k$ and DMR, indicating that the interaction between memories benefits visual quality and temporal consistency. When the causal mask in short-term enhancement is removed (Case 2), performance slightly drops, showing that causal modeling helps maintain motion continuity. Excluding the long-term memory as a conditioning signal (Case 3) results in the largest degradation among CME variants, proving the importance of

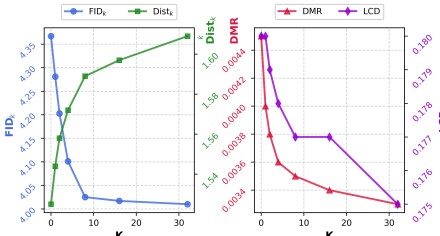

Figure 5: Ablation study for the impact of the number $K$.

Table 5: Ablation results of the TemMEGA model on CME and CMT components.

| Case | FID$_k \downarrow$ | Dist$_k \uparrow$ | DMR $\downarrow$ | LCD $\downarrow$ |
|------|------|------|------|------|
| Ours | 4.025 | 1.589 | 0.0035 | 0.177 |
| Case 1 | 4.238 | 1.552 | 0.0041 | 0.178 |
| Case 2 | 4.126 | 1.577 | 0.0038 | 0.178 |
| Case 3 | 4.367 | 1.525 | 0.0045 | 0.180 |
| Case 4 | 4.151 | 1.580 | 0.0037 | 0.178 |
| Case 5 | 4.851 | 1.425 | 0.0081 | 0.208 |
| Case 6 | 4.061 | 1.593 | 0.0036 | 0.177 |

long-term context for stable camera synthesis. Cases 4–6 investigate CMT. Injecting both memories through cross-attention (Case 4) achieves comparable but slightly worse performance than our design, suggesting that our fusion strategy is more effective. Using AdaIN for feature injection (Case 5) leads to a notable decline in all metrics, revealing that adaptive normalization is less suitable for temporal-memory fusion. Sequentially injecting long-term and then short-term memory (Case 6) performs close to the full model, demonstrating the robustness of the proposed memory arrangement.

# 5  Conclusion

In this paper, we introduce the TemMEGA framework, a novel approach for real-time camera movement synthesis tailored specifically for live dance performances. Unlike previous methods that rely on full-length dance videos, TemMEGA utilizes only current and past segments of dance and music, making it feasible for real-time application. Our approach leverages discrete camera tokenization, a consecutive memory encoder for capturing long- and short-term temporal dependencies, and a conditional masked transformer to generate camera movements dynamically. Experimental results on public datasets demonstrate that TemMEGA achieves state-of-the-art performance, validating its robustness and effectiveness in addressing the complexities of real-time camera movement synthesis for live dance contexts.

# 6  Acknowledgments

Our work is supported in part by the National Key R&D Program of China (No. 2023YFC3305600), the Joint Fund of Ministry of Education of China (8091B022149, 8091B02072404), National Natural Science Foundation of China (62132016, 62571412), and the Expert Workstation of Yunnan Province under Grant (202305AF150202).

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
