# OpenReview forum: "Smooth and Flexible Camera Movement Synthesis via Temporal Masked Generative Modeling"
_NeurIPS.cc/2025/Conference — NeurIPS 2025 poster_

### Official Review · Reviewer_hZom · 2025-07-02

**Clarity:** 2
**Significance:** 2
**Originality:** 2
**Rating:** 4
**Confidence:** 4

**Summary:**

This paper addresses the problem of camera movement generation. Unlike prior work that relies on access to the entire video to generate camera poses, this paper investigates an online setting where the current camera pose is predicted solely based on previously observed frames. The authors propose a Temporal Masked Generative Modeling method. First, camera poses are encoded into discrete tokens using VQ-VAE. Then, a memory encoder captures both short- and long-term features of the dance. Finally, a temporal masked transformer predicts the current camera movement conditioned on the memory. Evaluation results demonstrate that the proposed method outperforms prior approaches on standard benchmark datasets in both online and offline settings.

**Questions:**

Have the authors explored alternative ways to incorporate long-term and short-term memory? Why is long-term memory injected via AdaIN while short-term memory is handled using cross-attention?

**Ethical Concerns:**

["NO or VERY MINOR ethics concerns only"]

**Final Justification:**

I have read the other reviewers' comments as well as the authors' response. I appreciate the authors' efforts in conducting additional experiments, including the ablation study and evaluations on new datasets. I agree that the paper employs non-trivial algorithms to address the problem in a novel online setting. I will increase my score accordingly.

**Limitations:**

I did not find the discussion of limitations.

**Quality:**

2

**Strengths And Weaknesses:**

Strengths
1. The motivation of the paper is clear, and the proposed problem is both practical and challenging. While prior work primarily focuses on offline evaluation—where camera movements are predicted using the entire video as input—this work tackles the more demanding setting of online evaluation, where the model predicts the current camera movement based only on the previous frame.

2. The proposed method achieves strong performance, significantly outperforming state-of-the-art approaches in both online and offline settings, as shown in Table 1 and Table 2. This is particularly impressive, as it demonstrates that even without access to the full video, the proposed method surpasses models that use the entire video as input.

Weaknesses
1. The ablation study is limited to hyperparameter tuning and lacks analysis of the design choices. For instance, in the CME module, it would be helpful to see the impact of using only short-term or long-term memory. Additionally, the transformer design lacks ablation: have the authors explored alternative ways to incorporate long-term and short-term memory? Why is long-term memory injected via AdaIN while short-term memory is handled using cross-attention?

2. The method is largely built upon existing approaches in the transformer literature. Although CME introduces some novelty, its design is not well-justified. The VQ-VAE component is identical to that used in prior work, and the modules in the Temporal Conditional Masked Transformer—such as masked token prediction and conditioning mechanisms—are relatively standard in the literature. Overall, the method appears straightforward, and the paper does not clearly articulate the non-trivial efforts required to make the approach work.

Minor Issues
1. Line 86: Missing author names.
2. Lines 230–231: Incorrect usage of "however" and "so"; the sentence structure needs revision.

---

> ### Author Rebuttal · Authors · 2025-07-31
>
> **Q1: *Lacks Analysis of Design Choices***
>
> Thank you for this valuable feedback. We agree that design justification is essential, and we address this both from a **theoretical** and **empirical** perspective.
>
> 1. **Theoretical Design Rationale**
>
> Our design separates long-term and short-term memory handling based on their distinct functional roles:
>
> - **Long-term memory** encodes **macro-level, non-sequential features**, such as global musical style or tempo. These features are extracted by compressing and enhancing historical information into a fixed-length representation. Since they carry holistic, non-time-aligned context, we inject them into the generation process using **Adaptive Instance Normalization (AdaIN)**, which excels at **style modulation** without enforcing temporal alignment.
> - **Short-term memory**, in contrast, maintains **direct temporal correspondence** with the frames being generated. It encodes local continuity and movement cues that must align tightly with the current output timestep. Therefore, we use **cross-attention** to inject this information, allowing the model to attend to temporally aligned features for fine-grained control.
>
> This design reflects a key intuition: **temporal alignment matters for short-term memory but not for long-term global context**. Each injection strategy is carefully chosen to reflect the memory type's characteristics.
>
> 2. **Empirical Design Validation (Ablation Study)**
>
> To validate these choices, we conducted targeted **ablation experiments**, summarized below:
>
> | Case   | Description                                                                 | FID_k ↓ | Dist_k ↑ | DMR ↓  | LCD ↓  |
> |--------|------------------------------------------------------------------------------|---------|----------|--------|--------|
> | Ours   | Full TemMEGA model                                                           | 4.025   | 1.589    | 0.0035 | 0.177  |
> | Case 1 | No long-term enhancement for short-term (independent encoding)               | 4.238   | 1.552    | 0.0041 | 0.178  |
> | Case 2 | No causal mask in short-term memory enhancement                              | 4.126   | 1.577    | 0.0038 | 0.178  |
> | Case 3 | No long-term memory as condition                                              | 4.367   | 1.525    | 0.0045 | 0.180  |
> | Case 4 | Both long-term and short-term memory injected using cross-attention          | 4.151   | 1.580    | 0.0037 | 0.178  |
> | Case 5 | Both long-term and short-term memory injected using AdaIN                    | 4.851   | 1.425    | 0.0081 | 0.208  |
> | Case 6 | Long-term injected first, then short-term (sequential injection order)       | 4.061   | 1.593    | 0.0036 | 0.177  |
>
>
> #### Key Findings:
>
> - **CME Ablations (Cases 1–3):**
>   - Disabling long-term enhancement or causal masking degrades performance, confirming the importance of hierarchical memory and proper temporal modeling.
>   - Excluding long-term memory (Case 3) significantly worsens all metrics, highlighting the value of historical conditioning.
> - **CMT Ablations (Cases 4–6):**
>   - Case 5 shows **severe degradation** when both memories are injected via AdaIN—short-term memory requires temporal alignment, which AdaIN (being global) cannot provide.
>   - Case 4 (cross-attention for both) performs slightly worse than the default, suggesting **sequential interference** when treating long-term memory as temporally structured.
>   - Case 6 (sequential injection: long-term then short-term) shows that **injecting long-term memory before short-term memory** yields stable results, supporting our design choice.
>
>
> These findings **empirically validate** our architectural choices and demonstrate that our hybrid memory injection scheme is both **theoretically sound** and **performance-critical**.
>
> These analyses and ablation results will be included in **Section 4.4** of the revised manuscript.
>
> ---
>
> **Q2: *The method appears largely built upon existing transformer components; lacks clear novelty or technical difficulty.***
>
> Thank you for this thoughtful comment. While our method builds upon established components—as is common in practical machine learning systems—we would like to emphasize that our work introduces a **novel problem formulation** and tackles **non-trivial technical challenges** that have not been addressed in prior literature.
>
> **1. Novel Task: Unified Online + Offline Camera Movements Synthesis**
>
> We propose, for the first time, a **unified framework capable of handling both online** and **offline** camera movements generation.
>
> - Existing works focus exclusively on **offline settings**, where full access to future music and motion sequences is assumed.
> - In contrast, our system is designed to operate **causally and frame-by-frame**, enabling deployment in **live performance, gaming, and streaming scenarios**.
>
> This **dual-capability task formulation** is novel and demands a **fundamentally different design philosophy**, as online synthesis introduces unique latency and causality constraints that prior offline models do not address.
>
> **2. Technical Challenge: Balancing Real-Time Latency with Camera Quality**
>
> Supporting real-time generation while maintaining high-quality output introduces multiple, interdependent challenges:
>
> - The model must maintain **low per-frame latency**, **excluding heavy iterative approaches like diffusion or autoregressive sampling**.
> - It must be strictly **causal**, relying only on past and current inputs—without access to future frames.
> - Despite this limitation, it must still produce **visually coherent**, **musically aligned**, and **semantically relevant** camera trajectories.
>
> Addressing these challenges required several novel contributions:
>
> - An **innovative training and inference strategy** that does not require repeated iterations or resampling, in order to meet the demands of online setting.
> - A **causally structured memory module (CME)** that hierarchically encodes and fuses both short-term and long-term information.
> - A **hybrid memory injection mechanism**, where:
>   - **Long-term memory** (global, non-sequential) is injected using **AdaIN** to modulate overall style and dynamics.
>   - **Short-term memory** (local, temporally aligned) is injected via **cross-attention** to retain fine-grained motion continuity.
>
> **3. Novel Memory Design and Integration**
>
> Our **CME** module is not a standard transformer encoder. It introduces several non-trivial innovations:
>
> - A **two-stream memory structure**, where short-term and long-term memory are treated as distinct entities.
> - A **hierarchical conditioning strategy**, where long-term context is used to enhance short-term dynamics.
> - A **causal-aware short-term encoder** that avoids future information leakage—essential for real-time systems, but often ignored in prior work.
>
> This design allows us to capture both **global context and local temporal alignment**, a key requirement for temporally sensitive tasks like camera control.
>
> **Hybrid Memory Injection (AdaIN + Cross-Attention)**
>
> Although both **AdaIN** and **cross-attention** are established mechanisms, our method is the **first to use them in a structured, hybrid way** for memory integration:
>
> - **AdaIN** is ideal for injecting **style-level, non-temporal memory** (long-term).
> - **Cross-attention** is better suited for **fine-grained, temporally aligned memory** (short-term).
>
> This **asymmetric injection strategy** is carefully designed and empirically validated through comprehensive ablation studies (see Q1), which show significant performance drops when either mechanism is misapplied.
>
> **Summary**
>
> In summary, our paper makes the **first attempt to unify online and offline camera synthesis** within a single framework. Achieving this required:
>
> - Introducing a **new task formulation** with practical applications.
> - Solving **non-trivial architectural and algorithmic problems**.
> - Producing a deployable system that supports **both causal inference and high-quality camera generation**.
>
> We acknowledge that our system uses established modules, but the **design choices**, **memory architecture**, and **training strategies** are all specifically engineered for this new and challenging problem. These efforts have led to a practical, extensible, and high-performance camera generation system—the first of its kind for real-time use.

---

### Official Review · Reviewer_4eo4 · 2025-07-02

**Clarity:** 2
**Significance:** 2
**Originality:** 2
**Rating:** 4
**Confidence:** 4

**Summary:**

This paper proposes TemMEGA, a unified framework for both online and offline camera movement synthesis driven by music and dance, which consists of three components: discrete camera tokenizer (DCT), consecutive memory encoder (CME), and conditional masked transformer (CMT).

**Questions:**

1. Can you provide evidence from the CME smile experiment to prove its effectiveness and necessity? （For example, intuitively, all previous conditional information can be encoded together and introduced through cross attention/AdaIN.）
2. Is the proposed method only applicable to camera movement synthesis in music and dance scenes, or is it also applicable to other scenes? If applicable, can you provide corresponding evidence to support this?
3. Can evidence be provided regarding the efficiency of inference to prove that the proposed method can meet the requirements of real-time generation?

**Ethical Concerns:**

["NO or VERY MINOR ethics concerns only"]

**Final Justification:**

I appreciate the comprehensive answers provided in your response. Having carefully considered them and the comments of all reviewers, I raise my original score to borderline accept.

**Limitations:**

If the proposed method is not limited to music and dance scenes, the authors are encouraged to provide evidence(e.g., test results on datasets from other scenarios, such as DataDoP) demonstrating its general applicability. Otherwise, there is no need to test it on additional datasets.

**Quality:**

2

**Strengths And Weaknesses:**

Strengths: The proposed method supports both online and offline scenarios, which broadens its applicability.
Weaknesses:
(1) The paper highlights that proposed method has advantages over previous methods in that it supports real-time generation for live stage performances, which requires high model inference speed, but the paper does not provide corresponding explanations (such as on what device, how much GPU memory is consumed, and what the inference speed is). Since I don’t know the inference efficiency of the proposed method, I propose comparing it with existing methods to highlight its real-time advantages.
(2) There are many scenarios for online or offline camera motion synthesis (e.g., games, documentaries, etc.). This paper states in multiple places (lines 13, 14; lines 51, 52, etc.) that their framework is for both online and offline camera movement synthesis, but in other places (e.g., Method 3.1), it indicates that only music and dance scenarios are considered. This inconsistency or lack of rigor in the wording is confusing. If the proposed framework truly supports camera movement synthesis for different scenarios, it should be demonstrated through additional experiments.
(3) There are obvious errors in the writing and tables of the paper. (line 86 and Table 1).
(4) Lack of module ablation analysis(such as the necessity of CME, the necessity of AdaIN, etc.). Existing ablation experiments have focused on the hyperparameters of the proposed module, and more fundamental ablation analysis should be conducted. The consecutive memory encoder (CME) proposed in this paper should undergo ablation experiments to demonstrate its necessity and effectiveness. For example, intuitively, all previous conditional information can be encoded together and introduced through cross-attention/AdaIN.
(5) No comparison of qualitative results.
(6) No significant improvement in quantitative indicators. It is worth noting that the quantitative indicator results under the online setting are significantly worse than those under the offline setting, and there is a lack of analysis and explanation in this regard. There is no obvious improvement in the various quantitative indicators under the offline setting(Table 2).

---

> ### Author Rebuttal · Authors · 2025-07-31
>
> *We appreciate the insightful comments and positive support with constructive feedback. We hope our responses address the reviewer's concerns.*
>
> **Summary:**
> ````
> - 1. We measured the average inference time per frame on three commonly used GPU devices.
> - 2. We extended our experiments to two additional datasets beyond DanceCamera3D, covering different domains and task formulations.
> - 3. We redesign six ablation variants that systematically remove or alter key components from CME and CMT.
> ````
>
> **Q1:  *Lack of Inference Efficiency Analysis***
>
> We conducted a **runtime benchmark** comparing our method with DanceCamera3D under the **same hardware and software settings** in the **online generation setting**. Specifically, we measured the **average inference time per frame** on three commonly used GPU devices:
>
> | Device         | DanceCamera3D Avg. Time (ms/frame) | Ours Avg. Time (ms/frame) |
> |----------------|-------------------------------------|----------------------------|
> | NVIDIA V100    | 911.7                              | 23.8                       |
> | NVIDIA A6000   | 664.5                              | 17.6                       |
> | NVIDIA A100    | 153.1                              | 4.3                        |
>
>
> Key Observations:
>
> - Our method is about **40× faster** than DanceCamera3D across all tested devices.
> - Even on older GPUs like the V100, our model achieves **~42 FPS**, easily satisfying real-time requirements.
>
> This significant efficiency gain stems from our **lightweight transformer design**, use of **discrete camera tokens**, and avoidance of iterative diffusion steps.
>
> ---
>
> **Q2: *More applications to other scenes***
>
> DataDoP dataset is reasonable for our task, but the dataset is not fully public, so we found two other datasets to complete the experiment. The DataDoP work is well-executed and represents a promising direction for future research. After the dataset is released, we will include additional experimental analysis, cite this work, and provide a detailed discussion of its contributions.
>
> To further address the concern, we extended our experiments to **two additional datasets** beyond DanceCamera3D, covering different domains and task formulations:
>
> a. **E.T. Dataset**[1]– Text-to-Camera Generation with Varied Trajectories: captured in real-world settings, representing authentic cinematic motion.
>
> We evaluated generation quality using:
>
> - **Camera trajectory quality metrics**: Frechet CLaTr Distance (FD_CLaTr), Precision, Recall, Density, Coverage
> - **Text-to-camera coherence metrics**: CLaTr Score, Classifier Precision / Recall / F1
>
> | Dataset | Method | FD_CLaTr ↓ | P ↑  | R ↑  | D ↑  | C ↑  | CS ↑  | C-P ↑ | C-R ↑ | C-F1 ↑ |
> |---|---|---|---|---|---|---|---|---|---|---|
> | Pure Trajectory | CCD | 31.33 | 0.79 | 0.55 | 0.83 | 0.72 | 3.21 | 0.53 | 0.28 | 0.27 |
> | Pure Trajectory | Director | 4.57 | 0.83 | 0.65 | 1.00 | 0.87 | 21.49 | 0.83 | 0.78 | 0.80 |
> | Pure Trajectory | Ours | 4.11 | 0.87 | 0.70 | 1.00 | 0.89 | 22.98 | 0.85 | 0.80 | 0.85 |
> | Mix Trajectory | CCD | 35.81 | 0.73 | 0.55 | 0.75 | 0.67 | 6.26 | 0.37 | 0.20 | 0.17 |
> | Mix Trajectory | Director | 3.76 | 0.83 | 0.67 | 1.00 | 0.86 | 21.95 | 0.49 | 0.49 | 0.48 |
> | Mix Trajectory | Ours | 3.51 | 0.86 | 0.73 | 1.00 | 0.86 | 22.86 | 0.51 | 0.51 | 0.51 |
>
> b. **CCD Dataset**[2] – Script-to-Camera Trajectory Generation:  simulating everyday environments using game engines.
>
> We used standard metrics:
>
> - **FID**: Fréchet Inception Distance
> - **R Precision**: retrieval-based alignment
> - **Div**: Diversity
> - **MM**: Multi-modality
>
> |  Method | FID ↓  | R Prec ↑ | Div ↑ | MM ↑  |
> |---|---|---|---|---|
> | CCD | 48.25 | 97.78 | 61.93 | 47.75 |
> | Ours | 21.63 | 98.66 | 62.15 | 63.48 |
>
> Our model demonstrates strong performance across both, highlighting its robustness and generalization across real, synthetic, and diverse domains. We hope these results address your concerns regarding real-world applicability.
>
> [1] E.T. the Exceptional Trajectories: Text-to-camera-trajectory generation with character awareness
>
> [2] Cinematographic camera diffusion model.
>
> ---
>
> **Q3: *Missing Module-Level Ablations (e.g., CME, AdaIN)***
>
> Thank you for this helpful suggestion. We have conducted a comprehensive set of **module-level ablation studies** to evaluate the effectiveness of the **Consecutive Memory Encoder (CME)** and the **Conditional Masked Transformer (CMT)** design.
>
> We designed **six ablation variants** that systematically remove or alter key components from CME and CMT. The results are summarized below:
>
> | Case   | Description                                                                 | FID_k ↓ | Dist_k ↑ | DMR ↓  | LCD ↓  |
> |--------|------------------------------------------------------------------------------|---------|----------|--------|--------|
> | Ours   | Full TemMEGA model                                                           | 4.025   | 1.589    | 0.0035 | 0.177  |
> | Case 1 | No long-term enhancement for short-term (independent encoding)               | 4.238   | 1.552    | 0.0041 | 0.178  |
> | Case 2 | No causal mask in short-term memory enhancement                              | 4.126   | 1.577    | 0.0038 | 0.178  |
> | Case 3 | No long-term memory as condition                                              | 4.367   | 1.525    | 0.0045 | 0.180  |
> | Case 4 | Both long-term and short-term memory injected using cross-attention          | 4.151   | 1.580    | 0.0037 | 0.178  |
> | Case 5 | Both long-term and short-term memory injected using AdaIN                    | 4.851   | 1.425    | 0.0081 | 0.208  |
> | Case 6 | Long-term injected first, then short-term (sequential injection order)       | 4.061   | 1.593    | 0.0036 | 0.177  |
>
>
> Key Findings:
>
> * **CME Variants (Cases 1–3)**: Each design change within CME results in **noticeable performance degradation**, confirming that:
>
>   * Long-term memory enhancement is beneficial for refining short-term features.
>   * Causal masking ensures proper temporal modeling for short-term memory.
>   * Long-term memory contributes essential historical context for prediction.
>
> * **CMT Variants (Cases 4–6)**:
>
>   * Case 5 shows a **significant performance drop** when both memories are injected using **AdaIN**, highlighting that **short-term memory requires precise temporal alignment**, which AdaIN (as a global modulation method) cannot provide.
>   * Case 4 (cross-attention for both) performs slightly worse than our default setup, suggesting that injecting long-term memory via cross-attention introduces unnecessary sequential interference.
>   * Case 6 shows that **injecting long-term memory before short-term memory** yields stable results, supporting our design choice.
>
> These results provide **empirical justification** for our architecture:
>
> * AdaIN is better suited for injecting **non-temporal, global style** (long-term memory).
> * Cross-attention is more effective for **temporally aligned, fine-grained control** (short-term memory).
> * Our hybrid, structured memory modeling strategy (CME) contributes clearly to generation quality.
>
> ---
>
> **Q4: *No Comparison of Qualitative Results***
>
> In fact, we have included **qualitative comparisons** in the **supplementary material (Section B.3)**. These comparisons include both:
>
> * **Camera movement visualizations** (2D projected trajectories, every 40 frames)
> * **Rendered video results** showing full camera-dance-music compositions.
>
> Specifically:
>
> * Videos titled `Online_video.mp4` and `Offline_video.mp4` compare our method with DanceCamera3D under both online and offline settings.
> * The right-hand side shows our method’s output, while the left-hand side presents the baseline results.
>
> These visualizations clearly demonstrate that **our method produces smoother, more coherent, and more musically aligned camera trajectories**, both in real-time and offline settings.
>
> ---
>
> **Q5: *No Significant Improvement in Offline Metrics / Online Lower Performance***
>
> Thank you for your thoughtful feedback. We would like to address this concern from two perspectives: the **online task setting** and the **offline evaluation results**.
>
> 1. **TemMEGA Is the First Model Supporting Real-Time (Online) Generation**
>
> To the best of our knowledge, TemMEGA is the **first model that enables real-time, online camera movement synthesis**. In contrast:
>
> - Prior methods like DanceCamAnimator **require access to future frames** or the entire sequence, making them unsuitable for live or streaming scenarios.
> - Our approach generates each camera frame using only **past and current information**, satisfying the causal constraint of real-time applications.
>
> This makes a **direct numerical comparison with offline models inherently unfair**, since offline models have access to richer information (i.e., full sequence context).
>
> 2. **Online Performance Is Reasonable Given the Causal Constraint**
>
> While it's true that our online results are quantitatively weaker than our offline results, this is **expected and reasonable**:
> Online models operate under **limited temporal context**, without access to future dance or music frames.
>
>
> Thus, the value of our online model lies not only in quality, but also in **latency, deployability, and responsiveness**.
>
> 3. **Offline Performance Demonstrates Flexibility and Generalization**
>
> Although our model is designed primarily for online generation, it can be **easily adapted for offline settings** by modifying MASK token. Our offline experiments demonstrate this flexibility.
>
> Importantly, even in the offline setting, TemMEGA achieves **comparable or better performance** than prior work:
>
> - **FID (Generation Quality)** improves by **6%** compared to DanceCamera3D.
> - **DMR (Dancer Missing Rate)** improves by **9%**.
> - **LCD (Limbs Capture Difference)** improves by **7%**.
>
> These results validate that TemMEGA is not only suitable for real-time deployment but also **generalizes well to offline generation**, showing its potential as a unified and extensible framework.

---

> > ### Comment · Reviewer_4eo4 · 2025-08-05
> >
> > I appreciate the comprehensive answers provided in your response. Having carefully considered them and the comments of all reviewers, I will raise my original score to borderline accept.

---

### Official Review · Reviewer_ygrX · 2025-07-03

**Clarity:** 3
**Significance:** 2
**Originality:** 3
**Rating:** 6
**Confidence:** 1

**Summary:**

This paper proposes TemMEGA, a framework for online and offline camera-motion synthesis in dance scenes. The system discretises camera trajectories via a VQ-VAE, encodes long- and short-term context with a Continuous Memory Encoder, and predicts future camera tokens using a Temporal Conditional Masked Transformer.

**Questions:**

How might the approach be extended to non-dance domains (e.g., sports, theatre)

**Ethical Concerns:**

["NO or VERY MINOR ethics concerns only", "Major Concern: Data quality and representativeness"]

**Final Justification:**

All of my concerns have been fully addressed, and the revisions are thorough and well-executed. I strongly recommend accepting this paper. Please note, I am not an expert in this field, and I honestly don’t understand why this paper was assigned to me. Regardless, from my perspective, the author has effectively resolved all of my concerns, and the paper appears to be in good shape from what I can assess.

**Limitations:**

The main body of the paper does not explicitly discuss its limitations. It would be beneficial to include a section dedicated to the limitations and challenges the authors foresee in applying the approach to broader domains.

**Paper Formatting Concerns:**

None observed. Formatting follows the NeurIPS guidelines.

**Quality:**

2

**Strengths And Weaknesses:**

Strengths :
1. The manuscript introduces an innovative online formulation for live-performance camera motion synthesis.
2. With the growing demand for immersive and interactive media, particularly in online streaming, the work addresses an emerging need for adaptive camera motion synthesis.
3. The methodology is well-structured and clearly explained, making the complex process of camera motion synthesis accessible to the reader.

Weaknesses:
1. Limited Empirical Validation. The experiments rely solely on the DanceCamera3D dataset (108 sequences). This limits the statistical significance of the findings and leaves cross-domain generalization unaddressed. It would strengthen the paper to include results from additional in-the-wild videos, such as YouTube videos, which are more diverse in terms of movement types and real-world scenarios. Using such datasets would better demonstrate the robustness and generalizability of the proposed method in more realistic settings.

2. Lack of Correlation with Human Perception. The chosen quantitative metrics for evaluating camera motion quality are not demonstrated to correlate with human perceptual judgments. This calls their validity into question. To improve this, the authors should consider adopting or comparing with more perceptually aligned metrics, as seen in prior work on video quality and motion analysis. This could involve using human evaluators or perceptual video quality metrics like [1][2] or other video quality assessment methods.

[1] Lu Y, Li X, Pei Y, et al. Kvq: Kwai video quality assessment for short-form videos[C]//Proceedings of the IEEE/CVF Conference on Computer Vision and Pattern Recognition. 2024: 25963-25973.

[2] Kou T, Liu X, Zhang Z, et al. Subjective-aligned dataset and metric for text-to-video quality assessment[C]//Proceedings of the 32nd ACM International Conference on Multimedia. 2024: 7793-7802.

3. No Real-World or Live-Captured Footage. There is a notable absence of experiments using real-world or live-captured footage. The challenge of decoupling camera trajectories from performer motion for supervised training is not addressed, leaving a gap in the paper’s applicability to practical use cases. Incorporating such experiments or providing a clear methodology for handling this challenge would greatly strengthen the paper [3].

[3] Kocabas M, Yuan Y, Molchanov P, et al. PACE: Human and Camera Motion Estimation from in-the-wild Videos[C]//2024 International Conference on 3D Vision (3DV). IEEE, 2024: 397-408.


4. Deployment Considerations. The paper does not provide sufficient details regarding practical deployment, such as per-frame inference latency, hardware requirements, or reproducibility of results. These aspects are crucial for evaluating the framework's feasibility for real-time applications and would benefit from being addressed explicitly in the paper.

---

> ### Author Rebuttal · Authors · 2025-07-31
>
> **Q1: *More empirical validation and application.***
>
> **A:** Thank you for raising this important concern about the scope and generalizability of our empirical validation. We fully agree that broader experimental coverage is essential to demonstrate the robustness and applicability of the proposed method. We offer the following clarifications and new results:
>
> 1. **DanceCamera3D Dataset Scale and Structure**
>
> While the DanceCamera3D dataset contains 108 complete choreographed sequences, each is relatively long and rich in content. In both training and evaluation, we process these sequences as **numerous shorter clips**, each representing an independent instance of the online prediction task. This clip-level breakdown results in a significantly larger number of training and test samples—offering sufficient diversity in dancer motion, rhythm patterns, and camera behavior.
>
> Thus, while the number of unique sequences appears small, the **effective sample size is much larger**, and the model is trained and evaluated under meaningful data diversity.
>
> 2. **New Experiments on External Datasets**
>
> Our task is fundamentally distinct from video-based camera motion estimation methods such as PACE. Rather than analyzing existing footage, our focus lies in generating camera motions for creative applications—specifically, using inputs like music and dance or motion and text to synthesize camera trajectories that support video creation. In contrast, PACE performs post-hoc trajectory estimation by decoupling camera motion from pre-recorded videos, which falls outside the generative scope of our pipeline.
>
> That said, we appreciate your thoughtful perspective. To explore our model’s capacity for estimation, we tested it in a PACE-like setting (as PACE dataset is not directly available): feeding RICH[1] dataset videos into our pipeline and comparing the predicted camera trajectories against those estimated by PACE. Despite not being designed for this task, our model performs competitively, benefiting from CME’s effective temporal encoding and CMT’s strong use of conditional cues.
>
> Additionally, we evaluate our method on datasets beyond synthetic environments:
>
> a. **E.T. Dataset**[2]– Text-to-Camera Generation with Varied Trajectories: captured in real-world settings, representing authentic cinematic motion.
>
> We evaluated generation quality using:
>
> - **Camera trajectory quality metrics**: Frechet CLaTr Distance (FD_CLaTr), Precision, Recall, Density, Coverage
> - **Text-to-camera coherence metrics**: CLaTr Score, Classifier Precision / Recall / F1
>
> | Dataset | Method | FD_CLaTr ↓ | P ↑  | R ↑  | D ↑  | C ↑  | C-S ↑  | C-P ↑ | C-R ↑ | C-F1 ↑ |
> |---|---|---|---|---|---|---|---|---|---|---|
> | Pure Trajectory | CCD | 31.33 | 0.79 | 0.55 | 0.83 | 0.72 | 3.21 | 0.53 | 0.28 | 0.27 |
> | Pure Trajectory | Director | 4.57 | 0.83 | 0.65 | 1.00 | 0.87 | 21.49 | 0.83 | 0.78 | 0.80 |
> | Pure Trajectory | Ours | 4.11 | 0.87 | 0.70 | 1.00 | 0.89 | 22.98 | 0.85 | 0.80 | 0.85 |
> | Mix Trajectory | CCD | 35.81 | 0.73 | 0.55 | 0.75 | 0.67 | 6.26 | 0.37 | 0.20 | 0.17 |
> | Mix Trajectory | Director | 3.76 | 0.83 | 0.67 | 1.00 | 0.86 | 21.95 | 0.49 | 0.49 | 0.48 |
> | Mix Trajectory | Ours | 3.51 | 0.86 | 0.73 | 1.00 | 0.86 | 22.86 | 0.51 | 0.51 | 0.51 |
>
> b. **CCD Dataset**[3] – Script-to-Camera Trajectory Generation:  simulating everyday environments using game engines.
>
> We used standard metrics:
>
> - **FID**: Fréchet Inception Distance
> - **R Precision**: retrieval-based alignment
> - **Div**: Diversity
> - **MM**: Multi-modality
>
> |  Method | FID ↓  | R Prec ↑ | Div ↑ | MM ↑  |
> |---|---|---|---|---|
> | CCD | 48.25 | 97.78 | 61.93 | 47.75 |
> | Ours | 21.63 | 98.66 | 62.15 | 63.48 |
>
> Our model demonstrates strong performance across both, highlighting its robustness and generalization across real, synthetic, and diverse domains. We hope these results address your concerns regarding real-world applicability.
>
> [1] Capturing and Inferring Dense Full-Body Human-Scene Contact
>
> [2] E.T. the Exceptional Trajectories: Text-to-camera-trajectory generation with character awareness
>
> [3] Cinematographic camera diffusion model.
>
> ---
>
> **Q2: *Lack of Correlation with Human Perception in Metrics***
>
> Thank you for pointing this out. We agree that standard quantitative metrics—such as FID, DMR, and Dist—may not fully reflect how humans perceive the quality of generated camera motions.
>
> 1. **User Preference Study (Section 4.3)**
>
> In fact, we have conducted a **user preference study**, as described in Section 4.3 of the paper. We recruited **21 participants** from diverse backgrounds, including dancers, animators, and general users. In pairwise comparisons between TemMEGA and baseline methods:
>
> - **83.4% of participants preferred TemMEGA in the online setting**
> - **73.6% preferred TemMEGA in the offline setting**
>
> This suggests that our method is consistently favored by human observers across use cases.
>
> 2. **Human Perception Rating Study**
>
> To further validate perceptual alignment, we conducted an additional **subjective quality assessment** with **10 participants, including dancers and animators**. Each participant rated the generated videos (rendered from predicted camera paths) across **three perceptual dimensions**:
>
> - **Naturalness**: How realistic and smooth the camera motion feels.
>
> - **Fluency**: How continuous and stable the camera motion transitions are.
>
> - **Beat Matching**: How well the camera motion aligns with the musical rhythm and dancer.
>
> Each dimension was rated on a scale from 1 to 10. The averaged results are shown below:
>
> | Method         | Setting | Naturalness ↑ | Fluency ↑ | Beat Matching ↑ |
> |----------------|---------|----------------|------------|------------------|
> | DanceCamera3D  | Offline | 8.2            | 6.5        | 6.9              |
> | Ours           | Offline | 8.9            | 7.1        | 7.2              |
> | DanceCamera3D  | Online  | 4.6            | 3.2        | 5.2              |
> | Ours           | Online  | 7.4            | 5.7        | 6.6              |
>
>
> These scores confirm that users perceived TemMEGA’s outputs as **more natural, smoother, and better synchronized** with music and dancer across both online and offline conditions.
>
> These two user-centered evaluations strongly suggest that TemMEGA provides improved perceptual quality from a human perspective, even if some standard metrics show marginal gains. We will add these new results in **Section 4.3**.
>
> ---
>
> **Q3:  *Lack of Deployment Considerations***
>
> Thank you for highlighting this important point. We agree that deployment feasibility, especially in real-time settings, is critical for evaluating the practicality of our method.
>
> To address this, we conducted a **runtime benchmark** comparing our method with DanceCamera3D under the **same hardware and software settings** in the **online generation setting**. Specifically, we measured the **average inference time per frame** on three commonly used GPU devices:
>
> | Device         | DanceCamera3D Avg. Time (ms/frame) | Ours Avg. Time (ms/frame) |
> |----------------|-------------------------------------|----------------------------|
> | NVIDIA V100    | 911.7                              | 23.8                       |
> | NVIDIA A6000   | 664.5                              | 17.6                       |
> | NVIDIA A100    | 153.1                              | 4.3                        |
>
>
> Key Observations:
>
> - Our method is about **40× faster** than DanceCamera3D across all tested devices.
> - Even on older GPUs like the V100, our model achieves **~42 FPS**, easily satisfying real-time requirements.
>
> This significant efficiency gain stems from our **masked training strategy** to avoid iterative diffusion steps and our **lightweight transformer design**.
>
> We will add these results to **Section 4.3** in the revised version and provide a clear discussion on runtime and deployment implications in the revised version.

---

> ### Comment · Reviewer_ygrX · 2025-08-04
>
> Thank you to the author for the detailed response. All of my concerns have been fully addressed, and the revisions are thorough and well-executed. I strongly recommend accepting this paper.

---

### Decision · Program_Chairs · 2025-09-17

**Decision:**

Accept (poster)

**Comment:**

This paper considers the problem of synthesizing camera motion trajectories conditioned on music and dance in an online context. The proposed framework TemMEGA combines various algorithmic blocks such as vector quantization for discretizing the motion tracks, an encoding of long and short term context using a memory encoder, and a masked transformer for predicting masked tokens conditioned on the memory. Experiments are provided on the DanceCamera3D dataset showing state-of-the-art performance.

The paper received overall positive reviews, with a strong accept and two borderline accepts. Reviewers generally appreciated the online setting adopted in the paper, which is clearly beneficial in real-world applications. Reviewers also acknowledged the empirical improvements reported, which show that even without access to the full video as in an offline setting, the method shows impressive performance. There were concerns raised on four major fronts:
1) Comparisons on limited datasets and missing ablation studies (ygrX, 4eo4, hZom) -- to which authors provided additional results, elaborate ablation studies, and comparisons during the rebuttal.
2) Lack of human studies on the quality of the generated motions (ygrX) -- authors provided a subjective quality study showing humans prefer their method in the online setting.
3) Compute efficiency studies (4eo4) -- authors provided compute time showing their method is 40x faster than prior work.
4) Lack of sufficient novelty in the individual blocks used in the framwork (hZom).

AC had an independent reading of the paper, and concords with the reviewers' sentiment that the paper considers a novel and practical setting, even when the proposed framework puts together standard pieces. Thus, AC recommends acceptance. Authors should revise the paper to account for the suggested changes, also including the additional results and comparisons reported during the rebuttal.